# Self-Supervised Few-Shot Learning on Point Clouds

**Charu Sharma and Manohar Kaul**
Department of Computer Science & Engineering
Indian Institute of Technology Hyderabad, India
charusharma1991@gmail.com, mkaul@iith.ac.in

## Abstract

The increased availability of massive point clouds coupled with their utility in a wide variety of applications such as robotics, shape synthesis, and self-driving cars has attracted increased attention from both industry and academia. Recently, deep neural networks operating on labeled point clouds have shown promising results on supervised learning tasks like classification and segmentation. However, supervised learning leads to the cumbersome task of annotating the point clouds. To combat this problem, we propose two novel self-supervised pre-training tasks that encode a hierarchical partitioning of the point clouds using a cover-tree, where point cloud subsets lie within balls of varying radii at each level of the cover-tree. Furthermore, our self-supervised learning network is restricted to pre-train on the support set (comprising of scarce training examples) used to train the downstream network in a few-shot learning (FSL) setting. Finally, the fully-trained self-supervised network's point embeddings are input to the downstream task's network. We present a comprehensive empirical evaluation of our method on both downstream classification and segmentation tasks and show that supervised methods pre-trained with our self-supervised learning method significantly improve the accuracy of state-of-the-art methods. Additionally, our method also outperforms previous unsupervised methods in downstream classification tasks.

## 1 Introduction

Point clouds find utility in a wide range of applications from a diverse set of domains such as indoor navigation [1], self-driving vehicles [2], robotics [3], shape synthesis and modeling [4], to name a few. These applications require reliable 3D geometric features extracted from point clouds to detect objects or parts of objects. Rather than follow the traditional route of generating features from edge and corner detection methods [5, 6] or creating hand-crafted features based on domain knowledge and/or certain statistical properties [7, 8], recent methods focus on learning *generalized representations* which provide semantic features using labeled and unlabeled point cloud datasets.

As point clouds are *irregular* and *unordered*, there exist two broad categories of methods that learn representations from point clouds. First, methods that convert *raw point clouds* (irregular domain) to *volumetric voxel-grid representations* (regular domain), in order to use traditional convolutional architectures that learn from data belonging to regular domains (e.g., images and voxels). Second, methods that try to learn representations directly from raw unordered point clouds [9–12]. However, both these class of methods suffer from drawbacks. Namely, converting raw point clouds to voxels incurs a lot of additional memory and introduces unwanted *quantization artifacts* [16], while supervised learning techniques on both voxelized and raw point clouds suffer from the burden of manually annotating massive point cloud data.

To address these problems, research in developing *unsupervised* and *self-supervised* learning methods, relevant to a range of diverse domains, has recently gained a lot of traction. The state-of-the-art unsupervised or self-supervised methods on point clouds are mainly based on *generative adver-*

*sarial networks* (GANs) [13, 14, 17] or auto-encoders [18–21]. These methods work with raw point clouds or require either voxelization or 2D images of point clouds. And, some of the unsupervised networks depend on computing a *reconstruction error* using different similarity metrics such as the *Chamfer distance* and *Earth Mover's Distance* (EMD) [18]. Such metrics are computationally inefficient and require significant variability in the data for better results [14] [18].

Motivated by the aforementioned observations, we propose a self-supervised pre-training method to improve downstream learning tasks on point clouds in the *few-shot learning* (FSL) scenario. Given very limited training examples in the FSL setting, the goal of our self-supervised learning method is to boost sample complexity by designing supervised learning tasks using surrogate class labels extracted from smaller subsets of our point cloud. We begin by hierarchically organizing our point cloud data into *balls of varying radii* of a cover-tree $\mathcal{T}$ [22]. More precisely, the radius of the balls covering the point cloud reduces as the depth of the tree is traversed. The cover-tree $\mathcal{T}$ allows us to capture a *non-parametric representation* of the point cloud at multiple scales and helps decompose complex patterns in the point cloud to smaller and simpler ones. The cover-tree method has the added benefit of being able to deal more adaptively and effectively to sparse point clouds with highly variable point densities. We propose two novel self-supervised tasks based on the point cloud decomposition imposed by $\mathcal{T}$. Namely, we generate surrogate class labels based on the real-valued distance between each generated pair of balls on the same level of $\mathcal{T}$, to form a regression task. Similarly, for parent and child ball pairs that span consecutive

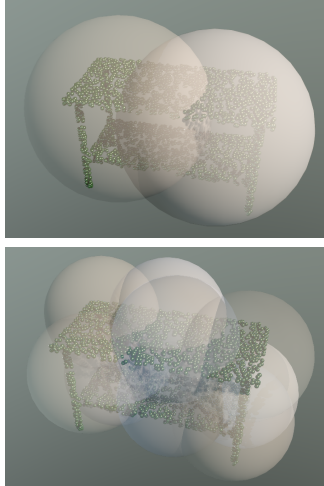

Figure 1: Coarse-grained (top) and finer-grained ball cover (bottom) of point cloud of *Table*.

levels of $\mathcal{T}$, we generate an integer label that is based on the quadrant of the parent ball in which the child ball's center lies. This, then allows us to design a classification task to predict the quadrant label. The rationale behind such a setup is to learn global *inter-ball* spatial relations between balls at the same level of $\mathcal{T}$ via the regression task and also learn local *intra-ball* spatial relations between parent and child balls, spanning consecutive levels, via the classification task. Our learning setup has the added advantage that these pretext tasks are learned at multiple levels of $\mathcal{T}$, thus learning point cloud representations at *multiple scales* and *levels of detail*. We argue that the combination of hierarchically decomposing the point clouds into balls of varying radii and the subsequent encoding of this hierarchy in our self-supervised learning tasks allows the pre-training to learn meaningful and robust representations for a wide variety of downstream tasks, especially in the restricted FSL setting.

**Our contributions:** To the best of our knowledge, we are the first to introduce a *self-supervised* pre-training method for learning tasks on point clouds in the FSL setup. (i) In an attempt to break away from existing raw point cloud or their voxelized representations, we propose the use of a metric space indexing data structure called a cover-tree, to represent a point cloud in a hierarchical, multi-scale, and non-parametric fashion (as shown in Figure 1). (ii) We generate surrogate class labels, that encode the hierarchical decomposition imposed by our cover-tree, from point cloud subsets in order to pose two novel self-supervised pretext tasks. (iii) We propose a novel neural network architecture for our self-supervised learning task that independently branches out for each learning task and whose combined losses are back propagated to our feature extractor during training to generate improved point embeddings for downstream tasks. (iv) Finally, we conduct extensive experiments to evaluate our self-supervised few-shot learning model on classification and segmentation tasks on both dense and sparse real-world datasets. Our empirical study reveals that downstream task's networks pre-trained with our self-supervised network's point embeddings significantly outperform state-of-the-art methods (both supervised and unsupervised) in the FSL setup.

## 2 Our Method

Our method proposes a cover-tree based hierarchical self-supervised learning approach in which we generate variable-sized subsets (as balls of varying radii per level in the cover-tree) along with self-determined class labels to perform self-supervised learning to improve the sample complexity of

the scarce training examples available to us in the *few-shot learning* (FSL) setting on point clouds. We begin by introducing the preliminaries and notations required for FSL setup on point clouds used in the rest of the paper (Section 2.1). We explain our preprocessing tasks by describing the cover-tree structure and how it indexes the points in a point cloud, followed by describing the generation of labels for self-supervised learning (Subsection 2.2.1). Our self-supervised tasks are explained in Subsection 2.2.2. We then describe the details of our model architecture for the self-supervised learning network followed by classification and segmentation network (Subsection 2.2.3).

## 2.1 Preliminaries and Notation

Let a *point cloud* be denoted by a set $P = \{x_1, \cdots, x_n\}$, where $x_i \in \mathbb{R}^d$, for $i = 1, \cdots, n$. Typically, $d$ is set to 3 to represent 3D points, but $d$ can exceed 3 to include extra information like color, surface normals, etc. Then a *labeled point cloud* is represented by an ordered pair $(P, y)$, where $P$ is a point cloud in a collection of point clouds $\mathscr{P}$ and $y$ is an integer class label that takes values in the set $Y = \{1, \cdots, K\}$.

**Few-Shot Learning on Point Clouds** We follow a few-shot learning setting similar to Garcia et. al. [23]. We randomly sample $K'$ classes from $Y$, where $K' \leq K$, followed by randomly sampling $m$ labeled point clouds from each of the $K'$ classes. Thus, we have a total of $mK'$ labeled point cloud examples which forms our *support set* $S$ and is used while training. We also generate a *query set* $Q$ of unseen examples for testing that is disjoint from the support set $S$, by picking an example from each one of the $K'$ classes. This setting is referred to as $m$-shot, $K'$-way learning.

## 2.2 Our Training

### 2.2.1 Preprocessing

To aid self-supervised learning[1], our method uses the cover-tree [22] , to define a hierarchical data partitioning of the points in a point cloud. To begin with, the *expansion constant* $\kappa$ of a dataset is defined as the smallest value such that every ball in the point cloud $P$ can be covered by $\kappa$ balls of radius $1/\epsilon$, where $\epsilon$ is referred to as the *base of the expansion constant*.

**Properties of a cover-tree [22]** Given a set of points in a point cloud $P$, the cover-tree $\mathcal{T}$ is a leveled tree where each level is associated with an integer label $i$, which decreases as the tree is descended starting from the root. Let $B[c, \epsilon^i] = \{p \in P \mid \|p - c\|_2 \leq \epsilon^i\}$ denote a closed $l_2$-ball centered at point $c$ of radius $\epsilon^i$ at level $i$ in the cover-tree $\mathcal{T}$. At the $i$-th level of $\mathcal{T}$ (except the root), we create a non-disjoint union of closed balls of radius $\epsilon^i$ called a *covering* that contains all the points in a point cloud $P$. Given a covering at level $i$ of $\mathcal{T}$, each covering ball's *center* is stored as a *node* at level $i$ of $\mathcal{T}$[2]. We denote the set of centers / nodes at level $i$ by $\mathcal{C}_i$.

**Remark 1** *As we descend the tree's levels, the radius of the balls reduce and hence we start to see* tighter *coverings that faithfully represent the underlying distribution of $P$'s points in a non-parametric fashion. Figure 1 shows an example of two levels of coverings.*

**Self-Supervised Label Generation** Here, we describe the way we generate surrogate labels for two self-supervised tasks that we describe later. Recall that $\mathcal{C}_i$ and $\mathcal{C}_{i-1}$ denote the set of centers / nodes at levels $i$ and $i - 1$ in $\mathcal{T}$, respectively. Let $c_{i,j}$ denote the center of the $j$-th ball in $C_i$. Additionally, let $\#Ch(c)$ be the total number of child nodes of center $c$ and $Ch_k(c)$ be the $k$-th child node of center $c$. With these definitions, we describe the two label generation tasks as follows.

Task 1: We form a set of all possible pairs of the centers in $C_i$, except self-pairs. This set of pairs is given by $\mathcal{S}^{(i)} = \{(c_{i,j}, c_{i,j'}) \mid 1 \leq j, j' \leq |C_i|, j \neq j'\}$. For each pair $(c_{i,j}, c_{i,j'}) \in \mathcal{S}^{(i)}$, we assign a real-valued class label which is just $\|c_{i,j} - c_{i,j'}\|_2$ (i.e., the $l_2$-norm distance between the two centers in the pair belonging to the same level $i$ in $\mathcal{T}$). Such pairs are generated for all levels, except the root node.

Task 2: We form a set of pairs between each parent node in $C_i$ with their respective child nodes in $C_{i-1}$, for all levels except the leaf nodes in $\mathcal{T}$. The set of such pairs for balls in levels $i$ and $i - 1$ is

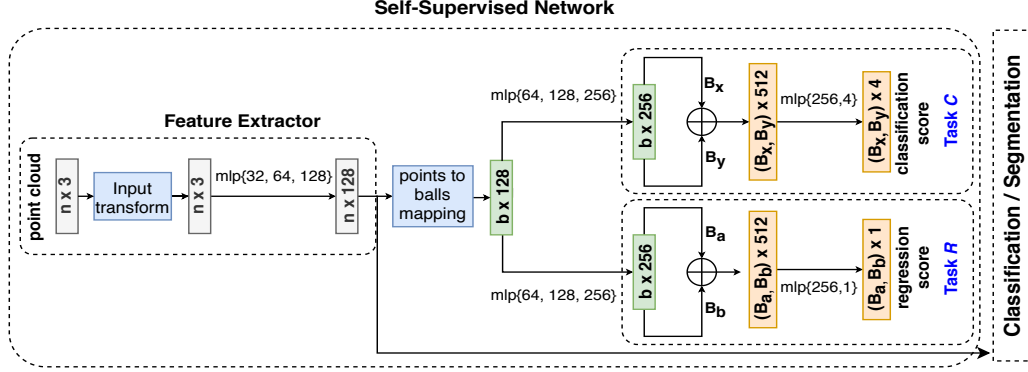

Figure 2: **Self-Supervised Deep Neural Network Architecture:** The architecture is used for pre-training using self-supervised labels for two independent tasks. **Feature Extractor:** The base network used for tasks during self-supervised learning can be used to extract point embeddings for further supervised training in a few-shot setting. Classification task $C$ and regression task $R$ are being trained in parallel using ball pairs taken from cover-tree $\mathcal{T}$ (described in Section 2). Here, $(B_a, B_b)$ and $(B_x, B_y)$ represent ball pair vectors. Classification or segmentation network is pre-trained with this model architecture.

given by $\mathcal{S}^{(i,i-1)} = \{(c_{i,j}, Ch_k(c_{i,j})) \mid 1 \leq j \leq |C_i|, 1 \leq k \leq \#Ch(c_{i,j})\}$. For each parent-child center pair $(p, c) \in \mathcal{S}^{(i,i-1)}$, we assign an integer class label from $\{1, \cdots, 4\}$ (representing *quadrants* of a ball in $\mathbb{R}^d$) depending on which quadrant of the ball centered on $p$ that the child center $c$ lies in.

### 2.2.2 Self-Supervised Learning Tasks

After outlining the label generation tasks, the self-supervised learning tasks are simply posed as: (i) Task 1: A *regression task* $R$, which is trained on pairs of balls from set $\mathcal{S}^{(i)}$ along with their real-valued distance labels to then learn and predict the $l_2$-norm distance between ball pairs from $C_i$ and (ii) Task 2: A *classification task* $C$, which is trained on pairs of balls from set $\mathcal{S}^{(i,i-1)}$ and their corresponding quadrant integer labels to then infer the quadrant labels given pairs from levels $C_i$ and $C_{i-1}$, where the pairs are restricted to valid parent-child pairs in $\mathcal{T}$.

### 2.2.3 Network Architecture

Our neural network architecture (Figure 2) consists of two main components: (a) the self-supervised learning network and (b) the downstream task as either a classification or segmentation network. Final point cloud embeddings from the trained self-supervised network are used to initialize the downstream task's network in FSL setting. We now explain each component in detail.

**Self-Supervised Network**   Our self-supervised network starts with a *feature extractor* that first normalizes the input point clouds in support set $S$, followed by passing them through three MLP layers with shared fully connected layers $(32, 64, 128)$ to arrive at 128-dimensional point vectors. For each ball $B$ in $\mathcal{T}$ (in input space), we construct a corresponding ball in feature space, by grouping in similar fashion, the point embeddings that represent the points in $B$. Then, a feature space ball is represented by a *ball vector*, which is the centroid of the point embeddings belonging to the ball. These ball vectors are then fed to two branches, one for each self-supervised task, i.e. $C$ and $R$, where both branches transform the ball vectors via three separate MLP layers with shared fully connected layers $(64, 128, 256)$. The ball pairs corresponding to each pretext task are represented by concatenating each ball's vector to get a $256 + 256 = 512$ dimensional vector. Finally, $C$ trains to classify the quadrant class label, while the regression task $R$ trains to predict the $l_2$-norm distance between balls in the ball pair. Note that losses from both tasks $C$ and $R$ are back-propagated to the feature extractor during training.

**Classification/ Segmentation**   At no point is the network jointly trained on both the self-supervised pre-training task and the downstream task, therefore any neural network (e.g., PointNet [10] and

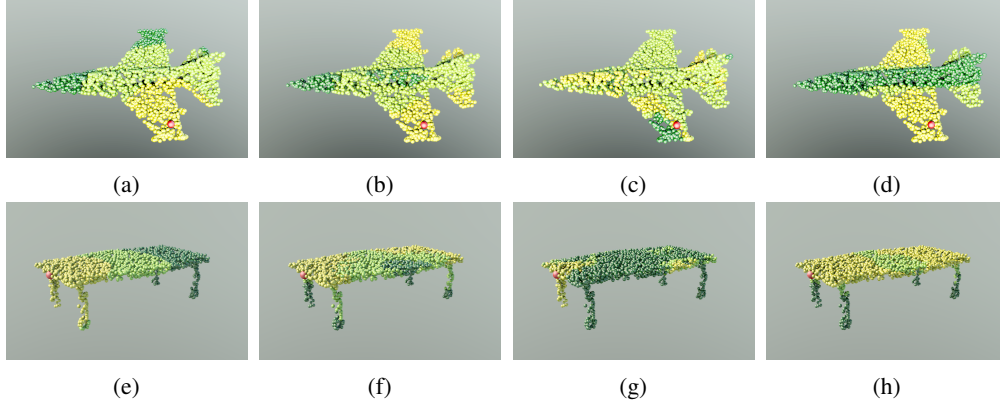

(a)  (b)  (c)  (d)

(e)  (f)  (g)  (h)

Figure 3: *Learned feature spaces* are visualized as the distance from the red point to the rest of the points (yellow: near, dark green: far) for *Aeroplane* and *Table* in (a, e) input 3D space. (b, f) feature space for DGCNN with random initialization. (c, g) feature space for DGCNN network pre-trained with VoxelSSL. And, (d, h) feature space of DGCNN pre-trained with our self-supervised model.

DGCNN [9]) capable of performing downstream tasks like classification or segmentation on point clouds can be initialized during training with the point embeddings outputted from our fully-trained self-supervised network's feature extractor.

We study the structure of the point cloud feature spaces compared to their input spaces via a heat map visualization as shown in Figure 3. As this is a FSL setup, the learning is not as pronounced as in a setting with training on abundant point cloud examples. Nevertheless, we observe that points in the original input space located on *semantically similar structures* of objects (e.g., on the wings of an aeroplane, on the legs of a table, etc.), despite initially having a large separation distance between them in the input space, eventually end up very close to one another in the final feature space produced by our method. In Figure 3a, points located on the same wing as the red point are marked yellow (close), while points on the other wing are marked green (far) in the original input space. But, in Figure 3d points on both wings (shown in yellow), being semantically similar portions, end up closer in the final feature space of our method. A similar phenomena can be observed in Figure 3 (second row), when we chose an extreme red point on the left corner of the table top. In contrast, the feature spaces of *DGCNN with random initialization* (Figures 3b, 3f) and *DGCNN pre-trained with VoxelSSL* (Figures 3c, 3g) do not exhibit such a lucid grouping of semantically similar parts.

## 3   Experiments

We conduct exhaustive experiments to study the efficacy of the point embeddings generated by our self-supervised pre-training method[3]. We achieve this by studying the effects of initializing downstream networks with our pre-trained point embeddings and measuring their performance in the FSL setting. For self-supervised and FSL experiments, we pick two real-world datasets (ModelNet40 [15] and Sydney[4]) for *3D shape classification* and for our segmentation related experiments, we conduct *part segmentation* on ShapeNet [24] and *semantic segmentation* on *Stanford Large-Scale 3D Indoor Spaces* (S3DIS) [25].

ModelNet40 is a standard point cloud classification dataset used by state-of-the-art methods, consisting of 40 common object categories containing a total of $12,311$ models with $1024$ points per model. Additionally, we picked Sydney as it is widely considered a hard dataset to classify on due to its sparsity. Sydney has $374$ models from $10$ classes with $100$ points in each model. ShapeNet contains $16,881$ 3D object point clouds from $16$ object categories which are annotated by $50$ part categories. S3DIS dataset contains 3D point clouds of $272$ rooms from $6$ areas in which each point is assigned to one of the $13$ semantic categories.

All our experiments follow the $m$-shot, $K'$-way setting. Here, $K'$ classes are randomly sampled from the dataset and for each class we sample $m$ random samples for support set $S$ to train the network. For query set $Q$, unseen samples are picked from each of the $K'$ classes.

Table 1: Classification results (accuracy %) on ModelNet40 and Sydney for few-shot learning setup. Bold represents the best result and underlined represents the second best.

| Method | ModelNet40 | | | | Sydney | | | |
|---|---|---|---|---|---|---|---|---|
| | 5-way | | 10-way | | 5-way | | 10-way | |
| | 10-shot | 20-shot | 10-shot | 20-shot | 10-shot | 20-shot | 10-shot | 20-shot |
| 3D-GAN | 55.80±10.68% | 65.80±9.90% | 40.25±6.49% | 48.35±5.59% | 54.20±4.57% | 58.80±5.75% | 36.0±6.20% | 45.25±7.86% |
| Latent-GAN | 41.60±16.91% | 46.20±19.68% | 32.90±9.16% | 25.45±9.90% | 64.50±6.59% | 79.80±3.37% | 50.45±2.97% | 62.50±5.07% |
| PointCapsNet | 42.30±17.37% | 53.0±18.72% | 38.0±14.30% | 27.15±14.86% | 59.44±6.34% | 70.50±4.84% | 44.10±1.95% | 60.25±4.87% |
| FoldingNet | 33.40±13.11% | 35.80±18.19% | 18.55±6.49% | 15.44±6.82% | 58.90±5.59% | 71.20±5.96% | 42.60±3.41% | 63.45±3.90% |
| PointNet++ | 38.53±15.98% | 42.39±14.18% | 23.05±6.97% | 18.80±5.41% | 79.89±6.76% | 84.99±5.25% | 55.35±2.23% | 63.35±2.83% |
| PointCNN | **65.41±8.92%** | 68.64±7.0% | 46.60±4.84% | 49.95±7.22% | 75.83±7.69% | 83.43±4.37% | 56.27±2.44% | 73.05±4.10% |
| PointNet | 51.97±12.17% | 57.81±15.45% | 46.60±13.54% | 35.20±15.25% | 74.16±7.27% | 82.18±5.06% | 51.35±1.28% | 58.30±2.64% |
| DGCNN | 31.60±8.97% | 40.80±14.60% | 19.85±6.45% | 16.85±4.83% | 58.30±6.58% | 76.70±7.47% | 48.05±8.20% | 76.10±3.57% |
| Voxel+DGCNN | 34.30±4.10% | 42.20±11.04% | 26.05±7.46% | 29.90±8.24% | 52.50±6.62% | 79.60±5.98% | 52.65±3.34% | 69.10±2.60% |
| Our+PointNet | 63.2±10.72% | **68.90±9.41%** | **49.15±6.09%** | 50.10±5.0% | 76.50±6.31% | 83.70±3.97% | 55.45±2.27% | 64.0±2.36% |
| Our+DGCNN | 60.0±8.87% | 65.70±8.37% | 48.50±5.63% | **53.0±4.08%** | **86.20±4.44%** | **90.90±2.54%** | **66.15±2.83%** | **81.50±2.25%** |

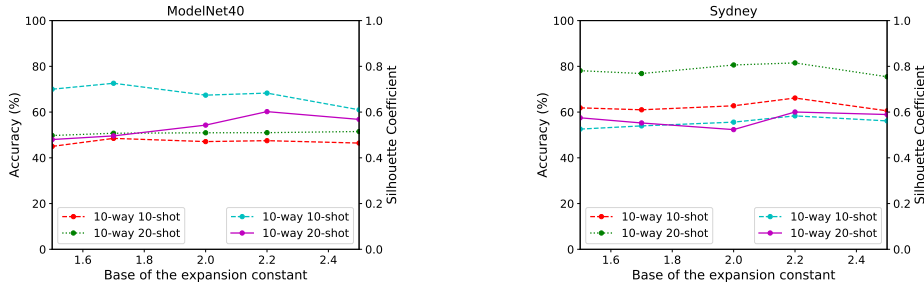

Figure 4: Comparison between *base of the expansion constant* $\epsilon$ (on $x$-axis) vs. *accuracy* ($y$-axis at left) and vs. *Silhouette coefficient* ($y$-axis at right) for ModelNet40 and Sydney datasets.

## 3.1 3D Object Classification

We show classification results with $K' \in \{5, 10\}$, and $m \in \{10, 20\}$ few-shot settings for support set $S$ during training on both the datasets. For testing, we pick 20 unseen samples from each of the $K'$ classes as a query set $Q$. Table 1 shows the FSL classification results on ModelNet40 and Sydney datasets for: i) unsupervised methods, ii) supervised methods, and iii) supervised methods pre-trained with self-supervised learning methods. All methods were trained and tested in the FSL setting. For the unsupervised methods (3DGAN, Latent-GAN, PointCapsNet, and FoldingNet), we train their network and assess the quality of their final embeddings on a classification task using linear SVM as a classifier. The linear SVM classification results are outlined in the first four rows of Table 1. Supervised methods (PointNet++, PointCNN, PointNet, and DGCNN) are trained with random weight initializations on the raw point clouds. Voxel+DGCNN indicates the DGCNN classifier initialized with weights from VoxelSSL [26] pre-training, while Our+PointNet and Our+DGCNN are the PointNet and DGCNN classifiers, initialized with our self-supervised method's base network point embeddings, respectively.

From the results, we observe that our method outperforms all the other methods in almost all the few-shot settings on both *dense* (ModelNet40 with 1024 points) and *sparse* (Sydney with 100 points) datasets. Note that existing point cloud classification methods train the network on ShapeNet, which is a large scale dataset with 2048 points in each point cloud and 16,881 shapes in total, before testing their model on ModelNet40. We do not use ShapeNet for our classification results, rather we train on the limited support set $S$ and test on the query set $Q$, corresponding to each dataset.

**Choice of Base of the Expansion Constant ($\epsilon$)** Recall that the *base of the expansion constant* $\epsilon$ controls the radius $\epsilon^i$ of a ball at level $i$ in the cover-tree $\mathcal{T}$. Varying $\epsilon$ allows us to study the effect of choosing tightly-packed (for smaller $\epsilon$) versus loosely-packed (for larger $\epsilon$) ball coverings. On one extreme, picking large balls results in coverings with massive overlaps that fail to properly generate unique surrogate labels for our self-supervised tasks, while on the other extreme, picking tiny balls results in too many balls with insufficient points in each ball to learn from. Therefore, it is important to choose the optimal $\epsilon$ via a grid-search and cross-validation. Figure 4 studies the effect of varying $\epsilon$

Table 2: Part Segmentation results (mIoU % on points) on ShapeNet dataset. Bold represents the best result and underlined represents the second best.

| | Mean | Aero | Bag | Cap | Car | Chair | Earphone | Guitar | Knife | Lamp | Laptop | Motor | Mug | Pistol | Rocket | Skate | Table |
|---|---|---|---|---|---|---|---|---|---|---|---|---|---|---|---|---|---|
| **10-way 10-shot** | | | | | | | | | | | | | | | | | |
| PointNet | 27.23 | 17.53 | 38.13 | _33.9_ | 9.93 | 27.60 | 35.84 | 11.26 | 29.57 | **42.63** | 27.88 | 14.0 | 31.2 | 19.33 | 20.5 | 34.67 | _41.78_ |
| PointNet++ | 26.43 | 16.31 | 33.08 | 32.05 | 7.16 | 30.83 | 34.0 | 12.22 | 29.81 | 39.09 | 28.73 | 12.66 | 30.33 | 17.8 | 21.44 | _35.76_ | 41.66 |
| PointCNN | 25.75 | 17.3 | 34.52 | 31.0 | 4.84 | 25.79 | 33.85 | 11.88 | 32.82 | 30.63 | 32.85 | 9.33 | _38.98_ | 21.77 | 20.69 | 29.4 | 36.35 |
| DGCNN | 25.58 | 17.7 | 33.28 | 31.08 | 6.47 | 27.81 | 34.42 | 11.11 | 30.41 | _42.31_ | 26.52 | 13.17 | 26.72 | 17.74 | 24.08 | 26.40 | 40.09 |
| VoxelSSL | 25.33 | 13.51 | 32.42 | 26.9 | 8.72 | 28.0 | _39.13_ | 8.66 | 29.08 | 38.28 | 27.38 | 12.7 | 30.65 | 18.8 | 24.02 | 27.09 | 39.98 |
| Our+PointNet | **36.81** | _24.59_ | **53.68** | 42.5 | _14.15_ | 42.33 | **44.75** | _26.92_ | 51.16 | 38.4 | _39.27_ | _15.82_ | **43.66** | _31.03_ | _31.58_ | **39.56** | **49.58** |
| Our+DGCNN | _34.68_ | **31.55** | _44.25_ | 31.61 | **15.7** | **49.67** | 36.43 | **29.93** | **58.65** | 25.5 | **44.43** | **17.53** | 32.69 | **37.19** | **32.11** | 30.49 | 37.22 |

on *classification accuracy* and the *Silhouette coefficient*[5] *of our point cloud embeddings*. DGCNN is pre-trained with our self-supervised network in a FSL setup with $K' = 10$ and $m \in \{10, 20\}$ for support set $S$ and 20 unseen samples from each of the $K'$ classes in the query set $Q$. We observe that $\epsilon = 2.2$ results in the best accuracy and cluster separation of point embeddings (measure via the Silhouette coefficient) for ModelNet40 and Sydney.

## 3.2 Ablation Study

Table 3 shows the results of our ablation study on two proposed self-supervised pretext tasks to study the contribution of each task individually and in unison. For support set $S$, $K' = 10$ and $m \in \{10, 20\}$ are fixed. The results clearly indicate a marked improvement when the pretext tasks are performed together. Although, we observe that learning with just the regression task (With only R) experiences better performance boosts com-

Table 3: Ablation study (accuracy %) on ModelNet40 and Sydney datasets for DGCNN with random init (without $C$/$R$) and pre-trained with our self-supervised tasks $C$ and $R$.

| Method | ModelNet40 | | Sydney | |
|---|---|---|---|---|
| | **10-way 10-shot** | **10-way 20-shot** | **10-way 10-shot** | **10-way 20-shot** |
| Without $C\&R$ | 19.85±6.45% | 16.85±4.83% | 48.05±8.20% | 76.10±3.57% |
| With only $C$ | 46.5±6.08% | 52.45±5.51% | 60.15±3.59% | 76.80±1.91% |
| With only $R$ | 47.0±6.41% | 50.70±4.99% | 62.75±2.83% | 80.60±2.25% |
| With $C + R$ | 48.50±5.63% | 53.0±4.08% | 66.15±2.32% | 81.50±2.59% |

pared to just the classification task (With only C). We attribute this phenomenon to the regression task's freedom to learn features globally across all ball pairs that belong to a single level of the cover-tree $\mathcal{T}$ (inter-ball learning), irregardless of the parent-child relationships, while the classification task is constrained to only local learning of child ball placements within a single parent ball (intra-ball learning), thus restricting its ability to learn more global features.

## 3.3 Part Segmentation

Our model is extended to perform part segmentation which is a fine-grained 3D recognition task. The task is to predict class labels for each point in a point cloud. We perform this task on ShapeNet which has 2048 points in each point cloud. We set $K' = 10$ and $m = 10$ for $S$. We follow the same architecture of part segmentation as mentioned in [9] for evaluation and provide point embeddings from the feature extractor of our trained self-supervised model.

We evaluate part segmentation for each object class by following a similar scheme to PointNet and DGCNN using the *Intersection-over-Union* (IoU) metric. We compute IoU for each object over all parts occurring in that object and the mean IoU (mIoU) for each of the 16 categories separately, by averaging over all the objects in that category. For each object category evaluation, we consider that shape category for query set $Q$ for testing and pick $K'$ classes from the remaining set of classes to obey the FSL setup. Our results are shown for each category in Table 2 for 10-way, 10-shot learning. Our method is compared against existing supervised networks i.e., PointNet, PointNet++, PointCNN, DGCNN and the latest self-supervised method VoxelSSL followed by DGCNN. We observe that DGCNN and PointNet pre-trained with our model outperform baselines in overall IoU and in majority of the categories of ShapeNet dataset in Table 2. In Figure 5, we compare our results visually with DGCNN (random initialization) and DGCNN pre-trained with VoxelSSL. From Figure 5, we can see

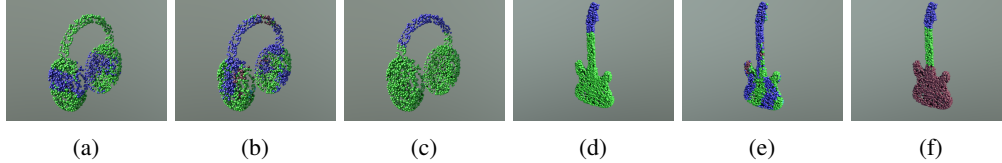

| (a) | (b) | (c) | (d) | (e) | (f) |

Figure 5: Part segmentation results are visualized for Shapenet dataset for *Earphone* and *Guitar*. For both the objects, (a, d) show DGCNN output, (b, e) represent DGCNN pre-trained with VoxelSSL and (c, f) show DGCNN pre-trained with our self-supervised method.

Table 4: Semantic Segmentation results (mIoU % and accuracy % on points) on S3DIS dataset in 5-way 10-shot setting. Bold represents the best result and underlined represents the second best.

| Test Area | Random Init | | VoxelSSL (pre-training) | | Ours (pre-training) | |
| | mIoU % | Acc % | mIoU % | Acc % | mIoU % | Acc % |
| --- | --- | --- | --- | --- | --- | --- |
| | | | **5-way 10-shot** | | | |
| Area 1 | $61.07 \pm 5.51$ % | $68.24 \pm 6.58$ % | $61.26 \pm 3.06$ % | $57.20 \pm 6.38$ % | **$61.64 \pm 3.11$%** | **$68.71 \pm 6.54$ %** |
| Area 2 | $55.94 \pm 4.48$ % | $61.52 \pm 4.36$ % | **$57.73 \pm 3.65$ %** | $60.45 \pm 2.80$ % | $\underline{56.43 \pm 6.20}$ % | **$64.67 \pm 4.07$ %** |
| Area 3 | $62.48 \pm 3.48$ % | $66.02 \pm 5.68$ % | $64.45 \pm 3.34$ % | $65.17 \pm 5.34$ % | **$64.87 \pm 6.43$ %** | **$67.07 \pm 7.25$ %** |
| Area 4 | $60.89 \pm 9.30$ % | $66.68 \pm 9.09$ % | $62.35 \pm 6.17$ % | $65.87 \pm 5.96$ % | **$62.90 \pm 7.14$ %** | **$71.60 \pm 3.92$ %** |
| Area 5 | $64.27 \pm 4.79$ % | $71.76 \pm 5.84$ % | **$68.06 \pm 2.77$ %** | $73.03 \pm 7.39$ % | $\underline{66.36 \pm 2.86}$ % | **$74.28 \pm 3.18$ %** |
| Area 6 | $63.48 \pm 4.88$ % | **$70.27 \pm 4.33$ %** | $60.65 \pm 3.22$ % | $65.82 \pm 4.90$ % | **$63.52 \pm 4.54$ %** | $66.88 \pm 4.85$ % |
| Mean | $61.36 \pm 5.41$ % | $67.41 \pm 5.98$ % | $62.42 \pm 3.70$ % | $64.59 \pm 5.46$ % | **$62.63 \pm 5.05$ %** | **$68.87 \pm 4.97$ %** |

that segmentation shown in our method (Fig. 5c, 5f) have far better results than VoxelSSL (Fig. 5b, 5e) and DGCNN with random initialization (Fig. 5a, 5d). For example, the guitar is clearly segmented into three parts in Fig. 5f and the *headband* is clearly separated from the *ear pads* of the headset in Fig. 5c.

## 3.4 Semantic Segmentation

In addition to part segmentation, we also demonstrate the efficacy of our point embeddings, via a semantic scene segmentation task. We extend our model to evaluate on S3DIS by following the same setting as [10], i.e., each room is split into blocks of area $1m \times 1m$ containing $4096$ sampled points and each point is represented by a 9D vector consisting of 3D coordinates, RGB color values and normalized spatial coordinates. For FSL, we set $K' = 5$ and $m = 10$ over 6 areas where the segmentation network is pre-trained on all the areas except one area (our support set $S$) at a time and we perform testing on the remaining area (our query set $Q$). We use the same metric mIoU% from part segmentation for each area and per-point classification accuracy as an evaluation criteria. The results from each area are averaged to get the mean accuracy and mIoU. Table 4 shows the results for DGCNN model with random initialization, pre-trained with VoxelSSL, and our self-supervised method. From the results, we observe that pre-training using our method improves mIoU and classification accuracy in majority of the cases with a maximum margin of nearly $3\%$ mIoU (Area 6) and $11\%$ accuracy (Area 1) over VoxelSSL pre-training and nearly $2\%$ mIoU (Areas 3, 4 and 5) and $5\%$ accuracy (Area 4) over DGCNN (random initialization).

## 4 Conclusion

In this work, we proposed to represent point clouds as a sequence of progressively finer *coverings*, which get finer as we descend the cover-tree $\mathcal{T}$. We then generated surrogate class labels from point cloud subsets, stored in balls of $\mathcal{T}$, that encoded the hierarchical structure imposed by $\mathcal{T}$ on the point clouds. These generated class-labels were then used in two novel pretext supervised training tasks to learn better point embeddings. From an empirical standpoint, we found our pre-training method substantially improved the performance of several state-of-the-art methods for downstream tasks in a FSL setup. An interesting future direction can involve pretext tasks that capture the geometry and spectral information in point clouds.

## Acknowledgements

We thank Jatin Chauhan from the Computer Science Dept. at IIT-H for his insights in the initial discussions. We would also like to thank all the reviewers for their feedback and suggestions. We are grateful to the authors of [9–14, 18, 19] for providing their source codes.

## Broader Impact

This study deals with self-supervised learning that improves sample complexity of point clouds which in turn aids better learning in a few-shot learning (FSL) setup; allowing for learning on very limited labeled point cloud examples.

Such a concept has far reaching positive consequences in industry and academia. For example, self-driving vehicles would now be able to train faster with scarcer point cloud samples for fewer objects and still detect or localize objects in 3D space efficiently, thus avoiding accidents caused by rare unseen events. Our method can also positively impact shape-based recognition like segmentation (semantically tagging the objects) for cities with limited number of samples for training, which can then be useful for numerous applications such as city planning, virtual tourism, and cultural heritage documentation, to name a few. Similar benefits can be imagined in the biomedical domain where training with our method might help identify and learn from rare organ disorders, when organs are represented as 3D point clouds. Our study has the potential to adversely impact people employed via crowd-sourcing sites like Amazon Mechanical Turk (MTurk), who manually review and annotate data.

## Footnotes

[1] We strictly adhere to self-supervised learning on the support set $S$ that is later used for FSL. [2] The terms *center* and *node* are used interchangeably in the context of a cover-tree.

[3]  Our code    [4]  Sydney Urban Objects Dataset

[5] The silhouette coefficient is a measure of how similar an object is to its own cluster (cohesion) compared to other clusters (separation). The higher, the better.

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
