[Supplementary Material]

# Self-Supervised Few-Shot Learning on Point Clouds

## Supplementary Material

**Charu Sharma and Manohar Kaul**
Department of Computer Science & Engineering
Indian Institute of Technology Hyderabad, India
charusharma1991@gmail.com, mkaul@iith.ac.in

## A    Additional Experimental Results

**Visualization of ball covers**    The cover-tree approach of using the balls to group the points in a point cloud is visualized in Figure 1. The visualization shows the process of considering balls shown as transparent spheres at different scales with different densities in a cover-tree. Fig 1a represents the top level (root) of cover-tree which covers the point cloud in a single ball i.e., at level $i$. Fig 1b and Fig 1c shows the balls at lower level with smaller radiuses as the tree is descended. Thus, we learn local features using balls at various levels with different packing densities.

### A.1    3D Object Classification

**Training**    This section provides the implementation details of our proposed self-supervised network. For each point cloud, we first scale it to a unit cube and then built a cover-tree with the base of the expansion constant $\epsilon = 2.0$ for all the classification and segmentation experiments. To generate self-supervised labels, we consider upto 3 levels of cover-tree and form possible ball pairs for both the self-supervised tasks $R$ and $C$. To train our self-supervised network, point clouds pass through our feature extractor which consists of three MLP layers $(32, 64, 128)$ and a shared fully connected layer. Similarly, for both the tasks, we use three MLP layers $(64, 128, 256)$ and shared fully connected layers in two separate branches. Dropout with keep probability of $0.5$ is used in fully connected layers. All the layers include batch normalization and LeakyReLu with slope $0.2$. We train our model with Adam optimizer with an initial learning rate of $0.001$ for 200 epochs and batch size 8. For downstream classification and segmentation tasks, we chose default parameters of DGCNN and PointNet for their training. We consider default parameters of all the baselines mentioned in their papers to train their respective networks.

**Effect of Point Cloud Density**    We investigate the robustness of our self-supervised method using point cloud density experiment on ModelNet40 dataset with 1024 points in original. We randomly pick input points during supervised training with $K' = 5$ and $m = 20$ for support set $S$ and 20 unseen samples from each of the $K'$ classes as a query set $Q$ during testing. The results are shown in Figure 2 in which we start with picking 128 points and go upto 1024 points for DGCNN, DGCNN pre-trained with VoxelNet and DGCNN and PointNet pre-trained with our self-supervised method. Figure 2 shows that even with very less number of points i.e., 128, 256, 512 etc. points, our method achieves comparable results and still outperform the DGCNN with random init and pre-trained with VoxelNet.

**T-SNE Visualization**    To verify the classification results, Fig 3 show T-SNE visualization of point cloud embeddings in feature space of two datasets (Sydney and ModelNet40) with 10 classes for DGCNN as classification network with random initialization, pre-trained with VoxelNet and our self-supervised network in a few-shot setup. We observe that DGCNN pre-trained with our method shows decent separation for both the datasets as compared to pre-training with VoxelNet which

(a) (A1)

(b) (A2)

(c) (A3)

Figure 1: Ball coverings of point cloud of object *Aeroplane* is visualized using cover-tree $\mathcal{T}$. Here, balls are taken from cover-tree to cover parts of the point cloud at different levels ($i$, $(i-1)$ and $(i-2)$) as the tree is descended for (A1, A2, A3), respectively.

Figure 2: Results with randomly picked points in a point cloud on ModelNet40 dataset in a 5-way 20-shot setting.

(a)  (b)  (c)

(d)  (e)  (f)

Figure 3: T-SNE visualization of point cloud classification for DGCNN network pre-trained with VoxelNet ((a), (d)), with random initialization ((b), (e)) and pre-trained with our self-supervised method ((c), (f)) in a few-shot setup for Sydney (first row) and ModelNet40 (second row) datasets.

is a self-supervised method and our main baseline. We also show better separation than DGCNN for Sydney dataset and nearly comparable separation to DGCNN with random initialization for ModelNet40 dataset in a few-shot learning setup.

**Heatmap Visualization**    We visualize the distance in original and final feature space as a heatmap in Fig. 4, 5, 6 and 7. It shows the distance between red point to all the other points (from yellow to dark green) in original 3D space in first row first column and final feature space for DGCNN with random init (first row second column), pre-trained with VoxelNet (second row first column) and our method (second row second column). Since this is a few-shot setup, the learning is not as good as it happens in a setting with all the point clouds but we observe that the parts of aeroplane in first and second rows of Fig. 4 such as both the wings are at the same distance and main body is far away from wings for our method whereas it differs for other methods. Similarly, in third and fourth row of Fig. 5 in table, the red point is on one of the legs of the table and all the other legs of the table are close to the red point in feature space (yellow) whereas table top is far away from the red point

in feature space (dark green) which is not the case with DGCNN with random initialization and DGCNN pre-trained with VoxelNet.

## A.2 Part Segmentation

We extend our model to perform part segmentation on ShapeNet dataset with $2048$ points in each point cloud. We evaluate part segmentation with $K' = \{5, 10\}$ and $m = \{5, 10, 20\}$ for support set $S$ and pick 20 samples for each of the $K'$ classes for query set $Q$. Table 1, 2, 3, 4, 5 and 6 show mean IoU (mIoU) for each of the 16 categories separately and their mean over all the categories. From Table 1, 2, 3, 4, we observe that DGCNN and PointNet pre-trained with our model outperform baselines in overall IoU and in majority of the categories while Table 5 and 6 shows either the best or the second best for our method in most of the cases. Our results for 10-way 10-shot setup is shown in main paper. Along with mIoU results, we also visualize part segmentation results for DGCNN with random init, pre-trained with VoxelNet and our method in Figure 9 and 8 for various object categories. We can see from the figures that segmentation shown in our method have far better results than VoxelNet and DGCNN with random initialization. However, in some cases we observe comparable results for both DGCNN with random init and pre-trained with our method. For example, the objects like Laptop and Bag have almost similar segmented parts for both DGCNN with random init and pre-trained with our method. On the other hand, our method produces properly segmented parts for more complex objects such as Car, Guitar, etc., as compared to the DGCNN with random initialization.

Table 1: Part Segmentation results (mIoU % on points) on ShapeNet dataset. Bold represents the best result and underlined represents the second best.

|  | Mean | Aero | Bag | Cap | Car | Chair | Earphone | Guitar | Knife |
|---|---|---|---|---|---|---|---|---|---|
|  | **5-way 5-shot** | | | | | | | | |
| PointNet | 26.35 | 18.31 | 28.80 | 30.12 | 6.94 | 28.60 | 36.06 | 11.27 | 30.31 |
| PointNet++ | 23.57 | 15.01 | 32.49 | 31.18 | 5.83 | 24.68 | 34.51 | 11.53 | 27.26 |
| PointCNN | 24.5 | 17.45 | 27.93 | 31.42 | 6.25 | 25.42 | 29.86 | 7.84 | 32.65 |
| DGCNN | 30.48 | 21.36 | 38.58 | 34.80 | 12.25 | 40.58 | 32.5 | 13.45 | 43.75 |
| VoxelNet | 22.51 | 17.01 | 30.41 | 31.38 | 8.55 | 21.78 | 27.49 | 8.88 | 27.01 |
| Our+PointNet | **33.67** | 20.02 | **40.18** | **39.97** | 12.64 | 41.16 | **37.82** | **21.86** | **59.29** |
| Our+DGCNN | 30.78 | **26.82** | 34.09 | 33.53 | **16.05** | **48.46** | 30.9 | 19.39 | 47.67 |

Table 2: Part Segmentation results (mIoU % on points) on ShapeNet dataset. Bold represents the best result and underlined represents the second best.

|  | Mean | Lamp | Laptop | Motor | Mug | Pistol | Rocket | Skate | Table |
|---|---|---|---|---|---|---|---|---|---|
|  | **5-way 5-shot** | | | | | | | | |
| PointNet | 26.35 | **38.32** | 28.70 | 11.25 | 27.33 | 17.65 | **39.04** | 29.85 | 39.04 |
| PointNet++ | 23.57 | 28.97 | 27.3 | 9.76 | 27.06 | 18.68 | 24.16 | 26.68 | 32.07 |
| PointCNN | 24.5 | 29.88 | 32.13 | 14.2 | 29.06 | 20.14 | 16.01 | 30.11 | 41.66 |
| DGCNN | 30.48 | 28.3 | **56.62** | 10.87 | 30.82 | 31.58 | 25.96 | 29.36 | 36.83 |
| VoxelNet | 22.51 | 27.42 | 25.0 | 11.33 | 7.76 | 17.83 | 18.71 | 24.22 | 34.95 |
| Our+PointNet | **33.67** | 33.79 | 45.73 | 14.55 | **35.92** | 29.14 | 27.85 | **33.82** | **45.05** |
| Our+DGCNN | 30.78 | 29.25 | 34.04 | **16.48** | 34.95 | **34.94** | 26.63 | 28.84 | 30.43 |

Table 3: Part Segmentation results (mIoU % on points) on ShapeNet dataset. Bold represents the best result and underlined represents the second best.

| | Mean | Aero | Bag | Cap | Car | Chair | Earphone | Guitar | Knife |
|---|---|---|---|---|---|---|---|---|---|
| | | | | **5-way 10-shot** | | | | | |
| PointNet | 25.9 | 15.02 | 34.37 | 32.96 | 8.40 | 27.76 | 34.15 | 9.32 | 28.2 |
| PointNet++ | 27.48 | _31.49_ | 32.16 | 27.41 | 9.06 | _45.42_ | 21.8 | 11.15 | 29.33 |
| PointCNN | 24.39 | 18.48 | 33.46 | 29.21 | 4.27 | 22.51 | 31.3 | 10.73 | 27.69 |
| DGCNN | 24.5 | 17.95 | 30.57 | 27.03 | 7.56 | 28.35 | _34.8_ | 9.53 | 27.65 |
| VoxelNet | 24.9 | 17.42 | 30.2 | 32.85 | 6.24 | 30.77 | **34.87** | 7.62 | 25.97 |
| Our+PointNet | **34.67** | 23.55 | **47.84** | **49.28** | _12.85_ | 38.99 | 32.26 | **31.16** | **51.69** |
| Our+DGCNN | _32.44_ | **35.75** | _37.69_ | _34.94_ | **16.94** | **46.74** | 33.88 | _15.75_ | _49.95_ |

Table 4: Part Segmentation results (mIoU % on points) on ShapeNet dataset. Bold represents the best result and underlined represents the second best.

| | Mean | Lamp | Laptop | Motor | Mug | Pistol | Rocket | Skate | Table |
|---|---|---|---|---|---|---|---|---|---|
| | | | | **5-way 10-shot** | | | | | |
| PointNet | 25.9 | 37.48 | 27.43 | 14.76 | 33.67 | 20.83 | 18.56 | _33.89_ | 37.54 |
| PointNet++ | 27.48 | _40.27_ | 29.01 | _19.26_ | **42.29** | 18.02 | _27.55_ | 27.05 | 28.41 |
| PointCNN | 24.39 | 30.16 | _31.25_ | 10.43 | 37.87 | 24.11 | 16.52 | 30.29 | 31.99 |
| DGCNN | 24.5 | 39.1 | 23.04 | 12.49 | 32.67 | 15.94 | 21.34 | 25.32 | 38.74 |
| VoxelNet | 24.9 | **40.82** | 23.31 | 13.01 | 24.87 | 19.42 | 24.02 | 27.09 | 39.98 |
| Our+PointNet | **34.67** | 32.91 | 29.94 | 16.82 | _42.2_ | **34.84** | **28.14** | 30.42 | **51.9** |
| Our+DGCNN | _32.44_ | 30.91 | **36.45** | **20.37** | 31.5 | _25.71_ | 23.53 | **37.85** | _41.05_ |

Table 5: Part Segmentation results (mIoU % on points) on ShapeNet dataset. Bold represents the best result and underlined represents the second best.

| | Mean | Aero | Bag | Cap | Car | Chair | Earphone | Guitar | Knife |
|---|---|---|---|---|---|---|---|---|---|
| | | | | **10-way 20-shot** | | | | | |
| PointNet | 27.4 | 18.07 | 37.94 | 32.85 | 8.72 | 29.85 | 34.02 | 10.6 | 29.18 |
| PointNet++ | **39.15** | 31.95 | 39.63 | **53.84** | **22.02** | _46.89_ | 33.66 | _26.9_ | 53.78 |
| PointCNN | 27.26 | 18.48 | 37.33 | 33.98 | 4.45 | 24.3 | 33.54 | 10.22 | 32.99 |
| DGCNN | _37.34_ | _37.13_ | 42.05 | _50.45_ | _17.2_ | **50.2** | **45.3** | **28.0** | _59.77_ |
| VoxelNet | 26.29 | 17.4 | 31.36 | 30.2 | 8.21 | 29.53 | 38.84 | 8.52 | 27.04 |
| Our+PointNet | 36.85 | 22.77 | **52.41** | 44.77 | 16.86 | 41.1 | 40.35 | 21.55 | 53.61 |
| Our+DGCNN | 36.91 | **37.72** | _47.34_ | 39.37 | 11.36 | 44.34 | _40.88_ | 25.77 | **61.71** |

Table 6: Part Segmentation results (mIoU % on points) on ShapeNet dataset. Bold represents the best result and underlined represents the second best.

| | Mean | Lamp | Laptop | Motor | Mug | Pistol | Rocket | Skate | Table |
|---|---|---|---|---|---|---|---|---|---|
| | | | | **10-way 20-shot** | | | | | |
| PointNet | 27.4 | _43.23_ | 28.56 | 14.8 | 34.02 | 17.6 | 19.29 | 35.79 | 43.87 |
| PointNet++ | **39.15** | 29.18 | 48.08 | _19.29_ | **42.99** | **52.39** | **43.8** | **40.39** | 41.68 |
| PointCNN | 27.26 | 35.18 | 31.83 | 12.14 | _42.54_ | 22.8 | 21.9 | 29.1 | 45.25 |
| DGCNN | _37.34_ | 35.57 | 25.22 | **20.69** | 40.4 | 37.21 | 26.93 | 35.7 | 45.54 |
| VoxelNet | 26.29 | **44.92** | 26.79 | 11.56 | 26.91 | 17.39 | 21.13 | 35.9 | 44.95 |
| Our+PointNet | 36.85 | 38.15 | _54.01_ | 19.16 | 34.46 | 34.67 | 27.54 | _39.14_ | **49.12** |
| Our+DGCNN | 36.91 | 36.94 | **56.24** | 15.82 | 34.48 | _41.94_ | 29.44 | 29.50 | _46.20_ |

Figure 4: **Learned feature spaces** are visualized as a distance between the red point to the rest of the points (yellow: near, dark green: far) for (A)*eroplane* and (L)*amp*. **A1, L1:** input $\mathbb{R}^3$ space. **A2, L2:** DGCNN with random initialization. **A3, L3:** DGCNN network pre-trained with VoxelNet. And, **A4, L4:** DGCNN pre-trained with our self-supervised model.

Figure 5: **Learned feature spaces** are visualized as a distance between the red point to the rest of the points (yellow: near, dark green: far) for (L)*aptop* and (T)*able*. **L1, T1:** input $\mathbb{R}^3$ space. **L2, T2:** DGCNN with random initialization. **L3, T3:** DGCNN network pre-trained with VoxelNet. And, **L4, T4:** DGCNN pre-trained with our self-supervised model.

Figure 6: **Learned feature spaces** are visualized as a distance between the red point to the rest of the points (yellow: near, dark green: far) for (E)*arphone* and (G)*uitar*. **E1, G1:** input $\mathbb{R}^3$ space. **E2, G2:** DGCNN with random initialization. **E3, G3:** DGCNN network pre-trained with VoxelNet. And, **E4, G4:** DGCNN pre-trained with our self-supervised model.

(M1)

(M2)

(M3)

(M4)

(M5)

(M6)

(M7)

(M8)

Figure 7: **Learned feature spaces** are visualized as a distance between the red point to the rest of the points (yellow: near, dark green: far) for (M)*otorbikes*. **M1, M5:** input $\mathbb{R}^3$ space. **M2, M6:** DGCNN with random initialization. **M3, M7:** DGCNN network pre-trained with VoxelNet. And, **M4, M8:** DGCNN pre-trained with our self-supervised model.

(T1)            (T2)            (T3)

(C1)            (C2)            (C3)

(C4)            (C5)            (C6)

(T4)            (T5)            (T6)

(B1)            (B2)            (B3)

Figure 8: Part segmentation results are visualized for ShapeNet dataset for (T)*ables*, (C)*ars* and (B)*ag*
. For each row, first column (T1, C1, C4, T4, B1) shows DGCNN output, second column (T2, C2, C5, T5, B2) represents DGCNN pre-trained with VoxelNet and third column (T3, C3, C6, T6, B3) shows DGCNN pre-trained with our self-supervised method.

Figure 9: Part segmentation results are visualized for ShapeNet dataset for (G)*uitar*, (P)*istol* and (L)*aptop*. For each row, first column (G1, P1, L1) shows DGCNN output, second column (G2, P2, L2) represents DGCNN pre-trained with VoxelNet and third column (G3, P3, L3) shows DGCNN pre-trained with our self-supervised method.