[Reviews · NeurIPS 2020]

Review 1

Summary and Contributions: This work proposes a two-stage method that first learns point embeddings from unlabeled data in a self-supervised fashion, which are then used for follow-up tasks including point cloud classification, part-segmentation and semantic segmentation. The paper introduces two self-supervised pretext tasks (allowing to pre-train the network for both regression and classification tasks) that are based on a novel hierarchical data structure for representing point clouds - i.e., the cover-tree.

Strengths: The paper introduces a novel application of cover-trees, a hierarchical representation to impose structure on 3D point clouds. Based on this representation, two new self-supervised learning tasks are introduced that allow learning point features without requiring labeled data. The efficiency of the pre-training and the contribution of the individual tasks is analyzed in a clear ablation study. Experiments on multiple downstream task show the improved scores over randomly initialized models.

Weaknesses: The number of parameters learned during the self-supervised tasks are relatively few - i.e., the number of weights of the MLP (32, 64, 128) - compared to the number of parameters of the downstream task models. In this setup, the self-supervision can only be used to generate stronger input features for a much larger subsequent model whose weights are still randomly initialized. The self-supervised regression task predicts the Euclidean distance between the centers of the sphere at the same level of the cover tree. What is the motivation behind this choice? How about predicting the rotation between two transformations of the same point cloud which might be a more challenging task than predicting the distance between points and therefore contribute more to learning improved features?

Correctness: l.32-35 The text suggests that the drawback of point cloud representations (as opposed to voxelized representations) is that they require labeled data. However, whether labeled training data is required for a method or not should be independent of the data representation. Can the authors further clarify what is meant here? Similar l.38-40, is there a specific reason why GANs or auto-encoder architectures would require voxelized data instead of point cloud representations?

Clarity: Generally, the paper is well written, the language is good and clear. Occasionally, it would be helpful to very briefly explain specific terms such as ‘sample complexity’, ‘query set’ for readers not yet familiar with the field of self-supervision and few-shot learning. (Similar l.48 - l.51). Some aspects are a bit unclear (see additional feedback below), most importantly l.13 ‘point embeddings are used to initialize the [...] network’ - what could mean is that the weights of the network are initialized using the *weights* obtained during pre-training. However Fig. 2, suggests that the learned point embeddings are used as input to the classification/segmentation networks. If this is the case, is the Feature Extractor fine-tuned during the downstream tasks?

Relation to Prior Work: l.18 ‘previous methods’ - how does this work compare to these previous methods? The paper does not explicitly discuss related work. While self-supervised learning and few-shot learning are indeed less explored in the 3D setting, there are some existing works: Choy, C., Park, J., Koltun, V.: Fully convolutional geometric features. In: ICCV (2019) Deng, H., Birdal, T., Ilic, S.: Ppf-foldnet: Unsupervised learning of rotation invariant 3D local descriptors. In: ECCV (2018) Zeng, A., Song, S., Nießner, M., Fisher, M., Xiao, J., Funkhouser, T.: 3DMatch: Learning local geometric descriptors from RGB-D reconstructions. In: CVPR (2017) A good source of additional related work is for example presented in the concurrent work of ‘PointContrast: Unsupervised Pre-training for 3D Point Cloud Understanding, ECCV 2020’

Reproducibility: No

Additional Feedback: l.56 ‘non-parametric’ - as opposed to which parametric representations? What about the radii of the spheres and the number of layers in the sphere? Why is the absence of parameters important? Specifically Remark 1 (l. 126), why is this relevant? l.142 How exactly is a quadrant of a sphere defined? E.g. is the spheres split along specific axis? What is d in R^d and which value did you use in your experiments? Is it the number of integer class labels / number of quadrants per sphere always set to 4 as suggested in l.142? l.185 ‘DGCNN pre-trained with VoxelNet’. Could the authors explain VoxelNet, how it is used to pre-train VoxelNet and if the trained weights are used for DGCNN (is it the same architecture?) or does VoxelNet represent point features that are used as input to the downstream task networks? How are the “ball vectors” obtained from the “point features” that fall within that ball? I.e., how is the blue box “points to ball mapping” in Fig.2 implemented? Somehow the point features need to be aggregated into ball features? Details: l.57 ‘recent methods’ - missing reference to these methods. l.107 typo: “Garcia et al.” --> “Garcia et. al.” ===================== After the reading the authors rebuttal and the discussion with the other reviewers my final rating leans towards the acceptance of the paper. However, the authors did not clearly address my concerns regarding the limited number of parameters of the MLP which are the only parameters optimized during the self-supervised learning phase. This can be insufficient as a generic feature extractor and should be further studied outside the few-shot-learning setup as also pointed out by other reviewers. At this point I am not convinced about the self-supervised learning part and these claims should play a lesser role in the paper.


Review 2

Summary and Contributions: The paper focuses on few-shot learning on 3D point cloud data and proposes a 3D self-supervised learning strategy to handle the scarce sets of labeled data. The paper proposes to represent a 3D point cloud with a cover tree data structure, which induces several pretext tasks for self-supervision purposes. The paper presents networks to extract useful point cloud features from these pretext tasks, and demonstrates the efficacy of the pre-training in both few shot object classification and few shot shape segmentation tasks.

Strengths: To my knowledge, this is the first attempt to leverage self-supervised learning for few shot learning tasks on 3D point cloud data, which is one of the main contribution of the paper. Also it is quite novel to leverage cover tree to define two complementary pretext tasks for self-supervised learning. The experimental evaluations are quite thorough, demonstrating the efficacy of the proposed method for few shot classification and segmentation. The paper is clearly written and technically sound. It will become a quite inspiring work for the community.

Weaknesses: Firstly, the paper does not have a clear discussion on its relation to and difference from existing works. Secondly, some technical details need further clarification. The cover tree is not uniquely defined for each point cloud and will the randomness cause trouble to the feature learning process? If not why? Maybe with some ablation study it will be easier to illustrate this. Thirdly, some experimental details are not very clear. For example, for ShapeNet part segmentation experiments, to generate the segmentation score for each category, how are the background categories picked up and how many experimental trials are done to obtain a single number?

Correctness: Yes as far as I can tell.

Clarity: The paper is quite clear.

Relation to Prior Work: The paper does not contain a related work section so the connection with and differences from previous works are not very clear.

Reproducibility: Yes

Additional Feedback: In general I like the idea behind the paper. Though there’re some presentation issues, I do not think they are hard to be addressed. One question in my mind, it seems self-supervised learning is not binded with few shot learning. Is the proposed self-supervised learning method also helpful for the normal supervised learning tasks? I also found a typo while reading: in the caption of Figure 1, Airplane should be Table. ==== Post Rebuttal ==== The authors have addressed most of my concerns in the rebuttal and I decided to keep my original score. One possible thing to fix is to tune down the claims regarding the self-supervised learning part since its efficacy is only evaluated on this FSL setting. Given the limited capacity of the backbone nets used in the self-supervised learning module, it is doubtful whether the pre-training could be helpful for a broader range of downstream tasks, which is usually something people value in SSL.


Review 3

Summary and Contributions: The paper proposes two self-supervised learning tasks based on cover tree to pre-train convolutional network weights. The tasks make use of the relationships of cover tree for a point cloud. One uses the distance between the nodes of the same level and the other uses the relative positions of the nodes at adjacent levels. The pre-trained model weights can serve as initialization for few-shot learning of point cloud classification and segmentation tasks. The experiments show encouraging results on those two tasks.

Strengths: The proposed approach can potentially improve the data label usage efficiency for 3D point cloud processing. Although self-supervised feature representation learning has been widely studied in image recognition tasks, few works have been done to address the similar problems in 3D point cloud recognition. The proposed approach is easy to understand and it may bring the trend of unsupervised featuring learning to the 3D domain.

Weaknesses: - The evaluation is done one few-shot learning on point cloud classification and segmentation. Table 1 shows that the results have big variance. Also, under the current evaluation protocol, PointNet is much better than PointNet++, which contradicts the full dataset classification results. Multiple validation and testing sets may be necessary to have a significant comparison between the methods. The same problem also arises in segmentation evaluation. The differences between ablation studies are within the error range in Table 4. - It is not clear whether the proposed method improves the results of PointNet++ and PointCNN. Those two achieved better results and different architectures than PointNet. It is important to know whether the proposed pre-training can work on different kinds of network structures. - It is not clear how good the pre-trained weights by self-supervising is compared to the fully supervised setting. ModelNet40 itself is not a very big dataset. Will the proposed method improve the feature learned on the full dataset? Can we learn the representation on a large dataset like ShapeNet and improve the state-of-the-art on ModelNet40? For feature learning, it is important to understand the gap between self-supervision and full supervision.

Correctness: Following the comments of the weakness, the evaluation strategy may not give convincing comparisons and it may not be comprehensive enough to show the full strength and weakness of the proposed method.

Clarity: This paper seems to be written in a hasty way. - Some long paragraphs are hard to pass on the first read. - It should be a point cloud of table instead of aeroplane in Table 1. - There are multiple missing latex figure references in the supplementary material.

Relation to Prior Work: The discussion is reasonable.

Reproducibility: Yes

Additional Feedback: One major concern is the evaluation protocol of few shot learning on point cloud. The current evaluation results in Table 1 have big variances, which even render some of the comparison insignificant. The rebuttal doesn't address this issue. Given this, I'm more inclined to keep my original score 5.


Review 4

Summary and Contributions: The paper proposes a new objective for self-supervised learning for point clouds. The idea is to divide the 3D space into balls of different sizes, which generates pseudo-labels that act as an auxiliary task for the self-supervised learner to solve. The learned representation is then validated on ModelNet, Sydney and ShapeNet, all of which have been adapted for the few-shot learning task. The authors claim contributions in: - First self-supervised method for the few shot learning task on point cloud. - A new formulation based on a hierarchical cover tree. - Multi-task learning architecture. - Experiments show that the SSLeared representation is useful. I agree on most points, except for the first one. I don't see why few-shot-learning is emphasized here. Just showing success on few-shot-learning is more of a limitation than an advantage, since showing a boost of SSL on fewer labeled data is easier than on a dataset with more labels.

Strengths: - Novel idea: Self-supervised learning for point cloud is still very unexplored. The hierarchical cover tree makes sense intuitively. - Extensive experimentation: 4 datasets, a handful of reimplemented baseline methods. A lot of experiments. - Relevance: point cloud representation learning is critical for many application in robotics, and is therefore very relevant to NeurIPS. - Paper is overall well written.

Weaknesses: - Problem statement: few-shot learning is somewhat uninteresting. As discussed before, I am not too fond of the idea of limiting the validation on few-shot learning. It implies that the representation learning is not helpful for larger datasets where more labels are available. It is rare in reality that you need to train a 10-way classifier, and there are only 10 samples for each of the 10 classes. I'd like to see this method working beyond few shot learning, otherwise, I don't believe that the problem setup itself is too interesting. However, given that there are few-shot learning papers out there, and considering all the other good points for this paper, I'll give it a favorable score. In the rebuttal, I'd like to understand if this sort of self-supervised learning brings a win for larger datasets. I know that the title of this paper is "few shot learning", but the few shot learning setup is just not very interesting. Since it's very easy to show wins of SSL when the supervised set is tiny.

Correctness: The claims and method seem correct and make sense. Some disagreements with the empirical methodology as discussed in "Weakensses".

Clarity: Yes, mostly. Some confusion by the title (see additional feedback).

Relation to Prior Work: Yes.

Reproducibility: Yes

Additional Feedback: - Actually, this point is very important. I find the title "self-supervised few-shot learning" very misleading. The goal of the paper is to use self-supervised learning to learn a representation that is to be used for few-shot learning, but not to use only a few samples to learn a representation via self-supervised learning. I'd suggest something like "self-supervised representation for few-shot learning".... - ln 34: Whether a method is supervised or not doesn't depend on whether it's voxelized or fully sparse. It's weird to assume any model that operates on raw point cloud is supervised. - ln 36, what's the difference between unsupervised and self-supervised learning in your opinion? - Introduction is too long and goes into a lot of details that's repeated later. Consider merging the section at ln 45 to methods. - Table 1.: Please indicate in the table which method is self-supervised + SVM, which ones are supervised, and which ones are finetuned. It's in the text, but the table has to be more or less self-contained. - ln 216: VoxelNet [a] is a well-known work for pointcloud, and its an encoding method and is unrelated to SSL. Not sure why the authors call [26] VoxelNet, since [26] does not call itself VoxelNet, seems confusing. a: https://arxiv.org/abs/1711.06396 **************************************************************************** POST REBUTTAL / REVIEWER DISCUSSION COMMENTS My main problem with this work was the FSL setup, which is not particularly interesting to me. The holy grail of self-supervised learning should be to improve results on benchmarks, and not to solve an artificially tougher problem. After some discussion with my fellow reviewers, who do not perceive this as a big issue, I'll upgrade my score to be inline with R1/R2. Best of luck!

[Author Response · NeurIPS 2020]

We thank all the reviewers for their diligence, appreciation of our work, and valuable comments / suggestions. Given the tight space constraint, we have done our best to address a majority of each reviewer's questions / comments.

**Reviewer 1: A1:** (The number of parameters..) We decided upon this architecture to be model-agnostic to downstream tasks and follow existing point cloud learning models (e.g. Wang, Yue, et al. "Dynamic graph CNN for learning on point clouds". ACM TOG 2019.) that similarly go from lower to higher number of parameters, progressively. **A2:** (The self-supervised regression..) We were inspired by the seminal work: C. Doersch, A. Gupta, and A. A. Efros. "Unsupervised visual representation learning by context prediction", ICCV 2015, which proposed a SSL task of *predicting the relative location of random image patches*. Your suggestion of rotations is well received and in fact we can even introduce different "local rotations" of the cover tree balls to make the pretext task even harder and more interesting. **A3:** (l.32-35 The text suggests..) Yes, this is a typo that distorts the meaning. We will rephrase it to convey that any supervised learning on BOTH voxel and raw point cloud representations require massive labeled point clouds, but voxelization has additional burdens of the memory overhead and introducing quantization artifacts. Thanks for pointing this out. **A4:** (l.38-40, is there a..) GANs and AEs can also work on raw PCs. We will omit this misleading statement. **A5:** (briefly explain specific terms...) We will add the explanation of *sample complexity*. Query set $Q$ is defined on l.110 − 112. **A6:** (most importantly l.13 point..) We will rewrite it as "the pre-trained point embeddings are input to the downstream network". **A7:** (l.56 'non-parametric'..) Parametric solid models like *Non-uniform rational basis spline* (NURBS) that best-fits a point cloud. Our intuition was that the surrogate labels generated from a non-parametric cover-tree like approach that could hierarchically data-partition the 3D input point cloud, by closely fitting to the point cloud's distribution, would result in: (i) a much more memory-efficient data structure (as opposed to voxel grids whose memory requirements grow cubically with resolution) and (ii) the labels would be more appropriate as they would avoid regions of "dead-space" (no points). **A8:** (l.142 quadrant ...) The sphere is in the 3D input space ($d = 3$) and quadrants are just the sphere divided by $xy$ and $yz$-planes passing through the sphere's center. **A9:** (How are the "ball vectors"..) The ball vector is the *barycenter* of the points present in the ball. The blue box in Fig. 2 just maps the point's indices to balls they are part of, to further compute the ball vectors.

**Reviewer 2: A1:** (The cover tree is not uniquely..) True, the cover tree is not unique. Finding a canonical and "optimal" covering of data points with balls of fixed diameter, where optimality can be defined as "least overlapping balls" or "least dead space of overall covering" etc., is NP-hard. The covertree was a good option as it uses a *fast and greedy construction approach* which takes *sub-linear time* and enjoys strong theoretical approximation guarantees. We also tried picking centers using a *kernel density estimation* (KDE) approach and found that it did not give us any improvements. An ablation study can certainly be added. **A2:** (for ShapeNet part segmentation:) For segmentation, we followed the experiment setup mentioned in Qi, Charles R., et al. "Pointnet: Deep learning on point sets for 3D classification and segmentation". CVPR 2017. Unlike images, our point cloud data had no background. We computed the IoU for each object over all parts occurring in that object and the mean IoU (mIoU) for each of the 16 categories separately. For each object category evaluation, we consider that shape category for query set $Q$ for testing and pick $K'$ classes from the remaining set of classes. We average our results over 10 query sets.

**Reviewer 3: A1:** (The evaluation is done on few-shot..) Given the scarcity of "FSL on point clouds" papers and that neither PointNet nor PointNet++ are developed with FSL in mind, it is not obvious that PointNet++ will always improve upon PointNet in the FSL setup. **A2:** (It is not clear whether the proposed..) Our SSL method is agnostic to the downstream network. We conducted a limited-setting experiment by SSL pre-training on ModelNet40 using our method and carrying out the classification task with Pointnet++ on ModelNet for a 10-way 10-shot FSL setting. We observed a boost in *classification accuracy* in Pointnet++ (without SSL) from 23.05% to 38.15% (with OurSSL). **A3:** (how good the pre-trained..) Given the limited time, we performed a classification task by pre-training with OurSSL on 4.2K (subset of ShapeNet) point clouds and then using DGCNN for the downstream full supervision task on ModelNet40. *Accuracy*: DGCNN (without SSL + Full supervision) = 91.73%, while DGCNN (with OurSSL + Full supervision) = 92.84%. We also conducted a similar classification task, by pretraining with OurSSL on the entire Sydney10 (sparse) dataset, followed by DGCNN in full supervision on the entire Sydney10 dataset. *Accuracy*: DGCNN (without SSL + Full supervision) = 92.3%, while DGCNN (with OurSSL + Full supervision) = 96.15%.

**Reviewer 4: A1:** (few-shot learning is somewhat uninteresting..) On the contrary, we find this work very interesting. Our aim is to build a hierarchical model-agnostic SSL model, so that downstream tasks like classification, segmentation (part and semantic) can use the pre-trained point cloud embeddings and adaptively learn extremely fast, with much fewer point cloud examples, thus improving the *transfer learning abilities* of FSL models. It is also widely accepted that FSL is a much more challenging setting because the downstream learner performs poorly, unless the data best covers the distribution of samples per class. FSL is a promising learning paradigm due to its ability to learn out of order distributions quickly with only a few samples. Please see our reply to Reviwer 3 A3 for more experimental results. **A2:** (ln 34: Whether a method ...) Please see our response to Reviewer 1 A3. **A3:** (Table 1.: Please indicate in the table..) Thanks, we will add this in cam-ready. **A4:** (ln 216: VoxelNet[a]..) Thanks for pointing this out. We will replace "VoxelNet" by "VoxelSSL" throughout our paper, as this is a self-supervised method on point clouds.

[Meta-Review · NeurIPS 2020]

Four knowledgeable referees reviewed the paper and overall they are in favor of acceptance. Key concerns include the large error bars in the tables, as well as the lack of experiments with more labeled data. Rebuttal generally does a good job at addressing the reviewers' concerns, including the second one above (but not the first one). All in all, I recommend acceptance, but the authors are strongly encouraged to address the reviewers' concerns in the final version of the paper.